# Eco-Concrete in High Temperatures

**DOI:** 10.3390/ma16124212

**Published:** 2023-06-06

**Authors:** Marcin Sundin, Hans Hedlund, Andrzej Cwirzen

**Affiliations:** Building Material Department of Civil, Environmental and Natural Resources Engineering, 97187 Lulea, Sweden; hans.hedlund@ltu.se (H.H.); andrzej.cwirzen@ltu.se (A.C.)

**Keywords:** eco-concrete, environmentally friendly, fire, elevated temperature

## Abstract

Concrete technology is becoming more and more sustainable and ecological following more extensive and focused research. The usage of industrial waste and by-products, such as steel ground granulated blast-furnace slag (GGBFS), mine tailing, fly ash, and recycled fibers, is a very important step toward a good transition of concrete into a “green” future and significant improvement in waste management in the world. However, there are also several known durability-related problems with some types of eco-concretes, including exposure to fire. The general mechanism occurring in fire and high-temperature scenarios is broadly known. There are many variables that weightily influence the performance of this material. This literature review has gathered information and results regarding more sustainable and fire-resistant binders, fire-resistant aggregates, and testing methods. Mixes that utilize industrial waste as a total or partial cement replacement have been consistently achieving favorable and frequently superior outcomes when compared to conventional ordinary Portland cement (OPC)-based mixes, especially at a temperature exposure up to 400 °C. However, the primary emphasis is placed on examining the impact of the matrix components, with less attention given to other factors such as sample treatment during and following exposure to high temperatures. Furthermore, there is a shortage of established standards that could be utilized in small-scale testing.

## 1. Introduction

There is a high global demand for sustainable and more environmentally friendly concretes characterized by a low CO_2_ footprint and long maintenance-free life span. Sufficient fire resistance is one of the properties that must be ensured. Concrete is a composite material composed of fine and coarse aggregates, water, active and inert fillers, and cementitious binder. Each of these components affect the ultimate performance, including the damage mechanism at high temperatures and when exposed to fire.

Common fire damage of concrete appears as discoloration, cracking, spalling, and decomposition of the binder matrix [1,2,3]. There are two commonly utilized methods for examining color alterations: (i) external and (ii) a cut-out section from a concrete element with visible aggregates [4]. The color change occurs due to the dehydration of the binder matrix, as shown in Table 1, and changes in the aggregate structure [5]. The presence of a siliceous type of aggregates exposed to temperatures from 300 °C to 600 °C causes a color change to red. At 600–900 °C, the color changes to white-gray and between 900 and 1000 °C to pale yellowish shades. The most noticeable color alteration (red) is caused by the presence of iron in GGBFS in the binder or aggregates [6] or siliceous riverbed aggregates [7]. However, calcium carbonate-rich aggregates turn into paler, whiteish tones following the calcination of CaCO_3_ [8]. The measurement of the color tone, saturation, and intensity give accurate and comparable data. These tests are based on measurements of the wavelength [9]. Najm et al. [10] reported that samples containing ceramic waste showed a color change to gray-beige tones at 200 °C and yellow at 300 °C. The presence of high-volume oil fuel ash, 50–70 wt% of cement, exposed to 800 °C caused a color change from pale gray to red [11]. Concrete mixes based on cements blended with GGBFS may have dark-blue shades which are related to the presence of sulfide ions in pores. It leads to GGBFS dissolution over alkali reactions [12]. Short et al. [13] and their data showed that the heat accelerates the oxidation process in specimens containing Portland cement and GGBFS. The core of the sample remained dark in contrast to the outer shell, which became lighter. It was observed that the blue-green color faded away before the appearance of red tones. The presence of the red color is associated with the reduction in the residual compressive strength of the specimen exposed to 350 °C. It could be used as a preliminary factor in the postfire assignment of damaged concrete. Visual changes in surfaces exposed to elevated temperatures enable an assessment of the extent of the damage.

The main source of crazing and cracking, which leads to delamination and spalling and then, afterward, to the destruction of the inner structure, is the thermal incompatibility between components of the binder matrix [14]. SEM images of eco-concrete mixes prepared by Salih et al. [15] showed that thermal cracks in the interfacial transition zone (ITZ) occurred at 800 °C. Kong and Sanjayan [16] stated that the presence of aggregates smaller than 10 mm could lead to more intense crack formation and spalling. The thermal expansion depended on the specimen size of the exposed element, as reported by Rickard et al. [17]. A larger size equated to a higher thermal expansion.

The amount of free and bound water has a significant impact on the dehydration. Thermogravimetric analysis (TGA) and magnetic resonance (NMR) of samples made of geopolymer showed that the majority of water is bound physically rather than chemically, as reported by Vickers et al. [18]. When the temperature rises, the water is released and shrinkage develops, pores shrink, and solid components, e.g., aggregates, expand. As a consequence, crystallization occurs and is followed by oxidization, sintering, and melting. Duxson et al. [19] reported that a metakaolin-based geopolymer upon exposure to a rising temperature showed a prolonged dehydration shrinkage due to the pore structure.

**Table 1 materials-16-04212-t001:** Water phase changes [20].

Temperature	Phase Changes
30–120 °C	Release of evaporable water, increase in vapor pressure.
>300 °C	Expulsion of chemically bound water.
>500 °C	Water capillary and gel pores water discharge. Pore volume rapid increase and changes in the pores system from isolated closed to interconnected network.

Spalling can be defined as a gradual peeling and scaling off of the external layers of concrete exposed to high temperatures. The mechanism remains largely unknown, but the increased vapor pressure created at elevated temperatures from entrapped water has been indicated as the leading factor [21]. The created internal pressure combined with thermal stress leads to explosive spalling. Interconnected pores facilitate the migration of gas or water from the internal to external part of an element. However, when the dynamic temperature changes, the built-up pressure and resulting stresses might exceed the tensile strength of the matrix [22]. Spalling has been directly linked with several building catastrophes [23,24,25]. The majority of published test results were obtained from “load free conditions” [26,27,28,29,30]. However, it was shown that mechanical loads, e.g., own-weight, permanent, and variable loads, snow, wind, earthquakes, and explosions, can have a significant impact too [21,31,32,33]. For example, explosive spalling occurred in samples exposed to 320–360 °C and uniaxial compressive stress [34]. Significant spalling also occurred on horizontal elements exposed to a high temperature and load of 15 MPa [35]. The slab spalling resistance threshold was found to be at 10 MPa [36].

Ref. [36] the duration of the fire-reached temperature and cooling period influence the fire resistance and thermal stability of concrete [37,38]. It is presumed and verified that the heat needs approximately one to two hours to penetrate a laboratory specimen to cause chemical and physical changes. It is associated with the thermal conductivity of the loaded element. The effect of elevated temperatures on the chemical composition of concrete is a crucial aspect of its performance under high-temperature exposure. Researchers have highlighted specific temperature limits, after which a noticeable alteration takes place within the matrix [39,40,41]. These changes can result in the destruction of bonds and, ultimately, lead to the collapse of the concrete structure. Therefore, it is essential to understand the underlying mechanisms behind these changes and develop strategies to mitigate their effect. 

OPC (ordinary Portland cement)-based concrete faces chemical decomposition at elevated temperatures, as shown in Table 2. It leads to the breakdown of the bonds between the cement particles and the aggregates. It causes the loss of strength and failure. The decomposition process starts at 200 °C. It involves the dehydration of calcium silicate hydrates (C-S-H) and the breakdown of calcium hydroxide Ca(OH)_2_ into calcium oxide (CaO). The absolute water evaporates. Other factors, such as the melting point of aggregates and the thermal expansion of concrete, also contribute to the degradation of OPC concrete at elevated temperatures.

There are several performance limitations of concrete based on OPC when subjected to high temperatures. This review paper will center on exploring alternatives to the usage of OPC and examine several environmentally friendly binding materials and their performance at high temperatures. It will also consider other critical factors such as an alkali-activated system, aggregates, and the significance of large-scale testing.

## 2. Fire Resistance of OPC Concretes with Supplementary Cementitious Materials (SCMs)

### 2.1. Ground Granulated Blast-Furnace Slag (GGBFS) Concrete

One of the commonly used construction materials in the building industry is steel [42]. During steel production, more precisely smelting ore, a GGBFS is obtained as a by-product. Steel GGBFS has also been used as a coarse and fine aggregate [43]. Ferrous GGBFS due to the presence of glass with a crystalline phase of silicate and calcium aluminosilicates is becoming more and more popular as a partial or full replacement of Portland cement in concrete. The first report regarding its hypothetical potential was produced by Feret in 1939 [44]. GGBFS can be used as a pozzolanic material that while intermixed with Portland cement, consumes part of the Portlandite and initiates the formation of calcium silicate hydrate (CSH) [45]. GGBFS, when used as a pozzolanic material, influences various properties of concretes, including exposure to high temperatures and fire. GGBFS concrete can undergo decomposition when exposed to fire, as shown in Table 3. The high temperatures cause a loss of strength and durability, and changes in the microstructure of the concrete. At temperatures between 200 °C and 400 °C, a new gel is formed which increases the density and strength of the concrete. However, at higher temperatures, calcium silicate hydrates (C-S-H) and calcium hydroxide Ca(OH)_2_ are present in the concrete dehydrate. The GGBFS particles themselves can also undergo changes and transform into crystalline phases. Studies showed that reference OPC concrete lost 14% of its residual compressive strength and mixes containing OPC/GGBFS blends only 4–5% when exposed to 800 °C. Furthermore, an OPC specimen showed cracking after exposure to 400 °C, while the OPC/GGBFS had no signs of any disintegration [46]. Samples that incorporated GGBFS and waste glass exhibited a lower deformation and reduced spalling. This can be attributed to their denser structure, with the melted glass pieces acting as internal locks to prevent crack propagation [47]. GGBFS has been used in prefabricated structural elements with improved fire resistance [48,49,50]. The properties were further enhanced in a mix containing 5 wt% of GGBFS and 5 wt% of fly ash. Bricks were exposed to 850 °C, 950 °C, and 1050 °C. Subsequently, the visual assessment highlighted only a slight color change, while no cracks and spalling or crazing were observed [51]. The presence of steel GGBFS in clay bricks along with zeolite and expanded perlite (EP) leads to increasing the bulk of the density. A compressive strength test has given values of 36.44 and 6.82 MPa, where the standard requirement for bricks is a minimum or equal to 7 MPa, according to the EN 771-3 [52].

Tests of concretes based on cement blends with 3 wt% and 40 wt% of BFS showed that the orientation of the crack has a major impact on the spalling scenario [53]. Inclined cracks to the heated surface should allow gases and vapor to be released, thus reducing spalling. Concretes containing a higher amount of GGBFS (43 wt% of Portland cement) showed less spalling when compared to the mix with 3 wt% of GGBFS. However, in biaxially loaded tests, the differences between the two mixes were insignificant. Additionally, it was found that the high permeability of the concrete did not have a strong effect on spalling under the given conditions. These findings suggest that the use of GGBFS as a replacement for cement in concrete mixes could be a promising approach in reducing spalling.

**Table 3 materials-16-04212-t003:** GGBFS concrete phase decomposition [54].

Temperature	Phase Changes
>200 °C	Evaporation of free, mass loss, better mechanical properties by temperature-induced gel formation.
>300 °C	Strength increase, formation of gel, decreasing porosity [55].
>500 °C	Strength loss due to gel crystallization, loss of strength performance.
>800 °C	The continued dehydration of calcium silicate hydrates (C-S-H) and the partial decomposition of carbonate.
>1200 °C	Decomposition of C-S-H, formation of ß-C2S and C3S, akermanite, merwinite, and gehlenite in the sodium sulfate-activated GGBFSs [56].

Poon et al. [57] compared the performance of normal and high-strength pozzolanic concretes containing silica fume, fly ash, and blast-furnace GGBFS exposed to a temperature of 800 °C. The threshold of good performance was set at 600 °C for mixes containing fly ash and GGBFS. Silica fume was highlighted as a component causing explosive spalling. It was concluded that a mix containing 40 wt% of GGBFS and 30 wt% of fly ash showed the best and optimum performance. Silica fume enhanced early compressive strength but did not affect the performance of specimens after the heat cycles (200 °C, 400 °C, 800 °C) [58]. Dissimilarities of concretes containing SCMs subjected to elevated temperatures were related to various sources of these materials. The performance of GGBFS-based concretes and their behavior at elevated temperatures was also evaluated by a machine learning methodology [59]. The temperature was not the most influential factor but rather the GGBFS/water and GGBFS/superplasticizer ratios. However, most analyzed data were obtained for temperatures < 400 °C. The influence of the ambient humidity led to an increase in the volume of components and further crack formation. Seleem et al. [60] came to the same conclusions while investigating the effect of elevated temperatures on cement blends containing varying amounts of GGBFS.

In general, GGBFS, but also concretes with fly ash (FA), seem to perform better in fire with less spalling and with narrower cracks. This trend was linked with the decreased amount of Ca(OH)_2_ [61]. The replacement of 40% cement by GGBFS gave the best results at temperatures < 600 °C.

### 2.2. Fly Ash (FA) Concrete

Low-calcium content fly ash is considered as a good pozzolanic material [62]. The results also showed that exposure to variable high temperatures for extended periods of time did not destabilize the microstructure of the binder matrix [63]. Chun et al. [64] reported that a concrete containing fly ash and a mix of fly ash and GGBFS performed better at a temperature of 800 °C, achieving 12.7 MPa (fly ash) and 13.5 MPa (fly ash and GGBFS) versus 9.9 MPa (pure cement) in a compressive strength test. However, over 90 days of recovery, the samples containing only pure cement gained the most strength, 18.5 MPa (pure cement) versus 15.3 MPa (fly ash) and 17.3 MPa (fly ash and GGBFS). A mix containing nanosilica and fly ash was investigated in a fire simulation by Mussa et al. [65]. The setup was arranged following an ISO 834 fire curve load over 120 min in a medium-scale furnace and a casted slab element (1850 mm × 1700 mm × 200 mm). It was reported that there was an inconsequential spalling at 680 °C present on 34.3% of the heated surface with a maximum depth of 23 mm. High-volume fly ash nanosilica (HVFANS) concrete was subjected to different levels of strain rates and varying temperatures of 25 °C, 400 °C, and 700 °C [66]. The obtained results indicate that the static and dynamic compressive strengths of HVFANS concrete are slightly weaker compared to plain concrete (PC) while tested at ambient temperatures. However, at elevated temperatures of 400 °C and 700 °C, the compressive strength of eco-concrete is greater than the PC reference sample. At ambient temperatures, the PC specimens yielded a result of 61.90 MPa, while the HVFANS specimens resulted in 58.71 MPa. When exposed to a temperature of 400 °C, the PC specimens yielded a result of 58.19 MPa, while the HVFANS specimens resulted in 64.55 MPa. At a temperature of 700 °C, the PC samples resulted in a value of 36.37 MPa, while the HVFANS resulted in 49.99 MPa. Referring to the present knowledge, certain temperature limits can be considered as indicators for detecting alterations in concrete containing fly ash, as shown in Table 4.

Fly ash-based concrete, under exposure to elevated temperatures, can undergo chemical decomposition. Between 200 and 400 °C, the new gel is formed, the density is increased, and the peak strength happens. The decomposition at higher temperatures results in the loss of strength and overall durability. The calcium silicate hydrates (C–S–H) and calcium hydroxide Ca(OH)_2_ present in the concrete dehydrates results in the formation of new phases such as gehlenite and anhydrite. Additionally, the fly ash particles themselves change and transform into crystalline phases such as mullite and hematite. These changes result in microstructural alterations and reduce the overall performance of the concrete in high temperature conditions.

### 2.3. Calcium Sulfoaluminate Concrete (CSA)

Calcium sulfoaluminate cement, due to lower CO_2_ emissions, is considered a suitable and ecological option for an OPC replacement [68]. The compressive stress in restrained concrete is increased due to an early age expansion. It subsequently results in the development of tensile during shrinkage. This reduced cracking is caused by shrinkage. The main phase in the hydration of CSA is ettringite [69]. Monosulfite, stratlingite, and alumina trihydrate [70] occur as additional phases. When exposed to high temperatures, the calcium sulfoaluminate (CSA) phases present in CSA-based concrete can undergo decomposition, as shown in Table 5. At higher temperatures, the decomposition of CSA phases can result in the loss of strength and overall durability. Between 200 and 230 °C, alumina trihydrate dehydroxylates. Monosulfite dehydrates above 450 °C. The decomposition reactions can result in the formation of new phases, such as gehlenite and anhydrite, and the dehydration of calcium sulfoaluminate hydrates and calcium sulphate hemihydrates.

This type of cement shows superior results in compressive strength tests, shrinkage behavior, gaining early strength, and resistance compared to OPC in ambient conditions [71]. The use of CSA instead of OPC resulted in a decrease of 38% in shrinkage. The compressive strength of the 28-day-old specimen was approximately 83 MPA, which is higher than the reference sample OPC, approximately 76 MPa. In another investigation, the specimens containing CSA were exposed to elevated temperatures for 15, 30, and 60 min [39]. The maximum temperature in the furnace reached 729 °C in 60 min. It has been noticed that CSA addition elongates the temperature growth. It occurs due to the ettringite endothermic reaction and dehydration of monosulfite. A nonspalling event was present, and the flexural and compressive strengths were improved for the mixes containing CSA. The OPC reference sample achieved 21.95 MPa in the compressive strength test, and the specimen containing CSA resulted in 18.22 MPa, respectively. 

Good insulation properties are essential in such solutions, as they help to prevent the spread of fire by reducing the amount of heat transferred from the fire-exposed side to the unexposed side of the structure. This can significantly increase the time available for evacuation and intervention by the fire department, potentially saving lives and reducing damage to the building. Moreover, a passive fire-protection solution made of materials with good insulation properties can reduce the cost and complexity of active fire-protection measures, such as sprinkler systems and fire-retardant coatings. Therefore, further research and development in this area is highly desirable. The investigation of mechanical and chemical changes occurring in CSA mixes at elevated temperatures and during fire has not been adequately examined. Most of the existing literature on CSA mixes concentrates on aspects such as their performance in normal conditions, manufacturing difficulties, such as a slow hydration rate caused by the presence of belite after 28 days [72], and the development of small, prefabricated components, such as bricks prepared with red mud and CSA cement [73].

## 3. Fire Resistance of Concretes Based on Alkali-Activated Binder Systems

Alkali-activated concrete, also commonly called geopolymer, was officially introduced in 1978 by French scientist Joseph Davidovits [74]. However, earlier traces lead to Ukraine, where in the late 1950s, Ukrainian scientist Glukhovsky discovered the high potential of an aluminosilicate-rich material mixed with a metal alkali solution [75]. This novel field of research was greatly impacted by the contributions of both specialists. The development and in-depth exploration of ecological and sustainable cementitious binders are yet to be achieved [76]. Industrial residues and their recycling [77], a shortage of raw materials [78], and a new approach for renewable sources of energy [79] have boosted a wave of advanced and cutting-edge research, tests, and applications for those new materials and solutions. Geopolymer concrete is characterized by a great potential for withstanding acid attacks [80], high temperature and fire resistance [16,81], and alkali–silica reactions [82]. The properties of geopolymer concrete depend on its compositions. 

The binder plays the main role due to its high chemical reactivity. Many studies in this field have reported that polymerizing materials rich with alumina-silicates, such as GGBFS [83], fly ash [84], rice husk [85], mine tailing [86], metakaolin [87] with alkaline solutions, aggregates, and water, are robust and resistant. As an activator solution, it has been widely studied and applied as a sodium/potassium silicate with sodium/potassium hydroxide [88]. They could be mixed with different ratios and molarity. With the right approach and methodology, the glass phase can be disturbed and activated by alkalis. NaOH and KOH solutions were applied to investigate the reactivity of GGBFS [89,90]. As a consequence of the destruction of internal bonds, it leads to a hydration process where calcium hydrate gel (CASAH) is obtained. The mechanism is very similar to the OPC hydration process with the difference being a higher amount of aluminum in the GGBFS case. CASAH is the most important phase for bond creation. Its amorphous microstructure is characterized by calcium interlayers with silicate tetrahedral sequences connected on two sides, supplemented with some aluminum-substituting chains of silicate tetrahedral [91]. In general, alkali-activated binder-based concrete is considered a dense material [92].

Elzaig ferrochrome slag (EFS)-based geopolymer concrete has shown a significantly improved fire resistance and less reduction in compressive strength and cracking or spalling than OPC concrete [93]. The main source of crack and delamination occurring over a heat load is thermal incompatibility between the components of the concrete matrix. However, research demonstrates that geopolymer concrete manifests remarkable thermal perseverance at the microscale [94]. Geopolymer concrete exposed to elevated temperatures showed less cracking, delamination, spalling, higher volume stability, and lower deformations [75,95,96]. The pore structure is more refined and connected, which affect water transport and vapor migration [97,98].

Fly ash geopolymer concrete has recently received a lot of attention due to its performance at elevated temperatures. Initial research has reported a slight increase in strength during exposure to a moderately low temperature of ~200 °C and relatively low loss of strength when exposed to higher temperatures compared to concrete specimens based on OPC [81]. Saker et al. [81] exposed a low-calcium fly ash-based geopolymer concrete to 400 °C, 650 °C, 800 °C, and 1000 °C [99,100]. The temperature gradient inside of the specimen was higher compared to OPC. The observed significant color difference after the exposure to high temperatures appeared useful for postfire visual assessment. In general, geopolymer concrete exhibited a higher rate of heat transfer compared to OPC concrete when exposed to fire, resulting in a lower temperature gradient within the geopolymer specimens. After exposure to temperatures above 650 °C, a significant color alteration occurred in the geopolymer concrete, ranging from brown to red. No spalling was observed, but the geopolymer specimen exhibited cracking after being subjected to temperatures of 800 °C and 1000 °C. The residual strength at 400 °C was 93% and 107%. In comparison, the OPC concrete specimen showed 90% of the residual strength. Geopolymer concrete experienced a decrease of 41% and 18% in strength at 600 °C, while OPC concrete had a 52% reduction in strength. The microstructure of the binder matrix remained stable but the incompatibility between the binder matrix and aggregates led to higher thermal strains. The compressive strength of the fly ash-based geopolymer concrete exposed to up to 800 °C was measured [29]. The results varied depending on the molarities of the NaOH activator, size of coarse aggregates, curing regimes, and different w/b. However, the geopolymer samples performed better at 600 °C and 800 °C compared to the OPC concretes. The concrete mix containing fly ash and activated by a 13 mol solution of NaOH performed the best in the compressive strength tests, as shown in Table 6. 

The enhanced performance of these blends was linked to the improved development of alumina-silicate networks during the process of geopolymerization. The better formation of this network is essential for improving the fire resistance of the geopolymer mixes, as it increases the stability of the material at high temperatures [101]. The microstructure of the material can better endure a high temperature exposure due to the high melting point of certain phases such as nepheline (NaAlSiO_4_), albite (NaAlSi_3_O_8_), and tridymite (SiO_2_).

Therefore, it is important to carefully consider the geopolymerization process when designing and manufacturing geopolymer mixes with improved fire-resistance properties. Although finer aggregates could enhance the formation of microcracks, which leads to the presence of bigger cracks and spalling phenomena [102]. Higher amounts of iron oxide in fly ash complemented the presence of red discolorations in samples after exposure to elevated temperatures [103]. Sodium and potassium activators incorporated in mixes containing fly ash and GGBFS were tested the most [30,103,104,105]. 

The Si/Al ratio tends to influence the microstructure and performance of geopolymer concrete. Geopolymers consisting of fly ash with high Si/Al (≥5) revealed an improved compressive strength and better volume stability at a temperature of 1000 °C compared to the geopolymer with the same binder but lower Si/Al ratio (<2) [103].

Another approach to enhancing the fire and high temperature properties of geopolymer concrete is the usage of selected filler and aggregates, which are more thermally resistant and stable. The application of a fine ceramic filler in a calcined kaolin clay and potassium silica solution geopolymer mix following Fuller’s gradation design (1 F/250) stood out with an undisturbed mechanical strength after a 1000 °C temperature exposure [106]. Thus, Nuaklong et al. [107] concluded that there is no significant difference regarding the spalling event and presence of granite waste and its moisture content. Basalt filler has improved all mechanical parameters in geopolymer concrete based on a metakaolin and potassium alkaline silicate solution at a temperature of 200 °C, 400 °C, 600 °C, 800 °C, and 1000 °C [108]. It could lead to the conclusion that by increasing the content of fine particles, the material might perform better in high temperature conditions. An interesting outcome was observed by adding basalt and glass fibers [109]. Color changes were explicit upon increasing the temperature. It could be very helpful for the postfire visual assignment of a damaged structure.

The fibers also minimized the mass loss and limited crack formation and spalling. Generally, there were no clear indications of enhanced mechanical properties. Geopolymer concrete, as a high-temperature and fire-resistant material with excellent thermal performance, chemical, and physical stability [110,111], could be applied in many ways, methods, and functions. However, the updates and revision of performance, alternative options, and upgrades should be carried out for its excellency, efficacy, and sustainability to promote them as practicable, user-friendly, and approachable for all types of facilities, infrastructures, and branches of constructions with a potential risk of fire and high-temperature events which could impact the safety, stability, and all hazards that could occur. Due to the absence of organic particles, it has been applied in products with an improved fire resistance [112,113,114,115,116]. The current state of the art indicates the development of heat-resistance geopolymers [112,117,118,119,120], binder formulas, and concrete mixes, which are suitable and applicable in the production of fire-proof rainscreens [121], small masonry elements [122], spray-applied fire-resistive material (SFRM) [122], and intumescent coatings [123,124]. A mix of metakaolin and GGBS filler applied in 10 mm thick geopolymer panels showed a great performance against a 1100 °C flame [125]. The temperature gradient between both sides of the panel was very different. The opposite side of the panel (not exposed to fire flame) reached only 350 °C after 35 min in the presence of heat. That feature makes it a perfect solution for a cover of fragile and nonfire-resistant elements.

Recent research and tests have proven the possible usage of a PFP system in the function of thermal resistance materials [126]. The merged alkali activation of fly ash with ladle GGBFS provides a proper start and foundation for using two very common industrial wastes in combination. Their inorganic nature, lack of ignition or combusting, multiresistance, and excellent performance to prolonged heat exposure in severe and rigorous fire testing bring to the table another approach based on synergy and complementing material characteristics. The flexural performance of reinforced geopolymer concrete in elevated temperatures was a research subject of Mathew and Joseph [127]. It was assumed that the appearance of the first crack under the applied load is mainly related to the rising temperatures. Increasing the temperature was manifesting the crack presence faster. The increased depth of the concrete cover helped to withstand the occurring deterioration. The specimen with a 20 mm thick cover started to crack at 66% of the applied load at a temperature of 800 °C. A 40 mm cover cracked at 75% under the same conditions. An advantageous network of fine cracks was reported on the surface of a specimen at 600 °C due to the presence of the 20% replacement of cement by volcanic ash [128]. It prevented spalling and it might provide a good basis for self-healing research. A study conducted by Razak et al. [129] evaluated the fire performance of geopolymer concrete based on fly ash at 500 °C and 1200 °C over a period of 120 min. The alkali-activated mix showed significantly higher results than the OPC reference samples. The visible improvement in the residual compressive strength occurred at 500 °C (145% of initial strength) due to geopolymerization, while the OPC specimen lost almost 50% of its resistance.

The increased density of geopolymer concrete prevent spalling and cracking events. According to the current state of the art, the data obtained are reliable enough to demonstrate significant similarities in the performance and properties of alkali-activated materials and ceramics, which indicates enhanced resistance to high temperatures and fire compared to OPC concrete.

## 4. Test Setups

Large-scale fire tests are considered to be the most reliable method of determining the fire resistance of materials and structures, as they provide a realistic scenario of a fire situation. However, they come at a high cost and require significant resources, making them challenging to conduct on a regular basis. These tests involve subjecting a full-scale structural element or system to a fire scenario that closely resembles real-world conditions, including fire load, temperature, and duration. Due to the expense and complexity of large-scale tests, most of the tests are conducted in specialized laboratories or institutions with accreditation for such testing. There are few standards applied in some countries. A Euroclass system was introduced in all European Union countries by the European Committee for Standardization, CEN [130]. Most scientists conduct small-scale experiments in laboratories using muffle furnaces, infrared thermometers, thermal cameras, thermocouple setups, and standard building materials equipment [131]. This could result in ambiguity and a limited ability to reproduce experiments and validate collected data. Therefore, it is crucial to implement a quality assurance process in laboratory tests and setups.

### 4.1. Small-Scale Tests

Small-scale fire tests are commonly used by researchers to investigate the fire resistance of concrete materials due to their low cost and easy logistics. These tests involve subjecting small concrete specimens to elevated temperatures and measuring their response. While small-scale tests have limitations, such as the inability to replicate the complex behavior of large-scale structures, they provide a valuable starting point for exploring the fire resistance of concrete materials. One of the primary distinctions is that the specimens are subjected to elevated temperatures in an electric furnace, rather than actual fire or flames [132]. However, it is important to note that small-scale tests should not be the sole basis for assessing the fire resistance of concrete structures and that further large-scale testing is necessary to ensure reliable data and the highest safety standards. After curing in the selected conditions, the samples undergo a series of measurements and observations to gather data on their physical and mechanical properties. This includes weighing the samples to determine their mass, measuring their dimensions to assess their size and shape, and photographing them to document their appearance and any visible defects or anomalies [133,134]. These measurements and observations provide valuable information on the quality and consistency of the concrete and can help identify any issues or areas for improvement in the production process. Additionally, the data collected from these tests can be used to validate models and simulations of concrete behavior, and to inform the development of new and improved materials for construction applications [135,136]. The accuracy and reliability of the data obtained from testing concrete specimens at elevated temperatures are highly dependent on the heating regime. Factors such as the speed and duration of the temperature load, placement of thermocouples, and methods of cooling specimens can have a significant impact on the test results. In addition, the ambient humidity level and sample size should also be taken into consideration to ensure an accurate and reliable outcome. Therefore, it is essential to carefully control these factors to obtain valid and reproducible data, which can be used to improve the fire safety of concrete structures. Zhao et al. [131] proposed a modified furnace, which was equipped with a circular opening to test vertical exposure on cylindrical samples. Thermocouples were installed on the samples to monitor the temperature changes. Thus, Anisah et al. [137] performed the high temperature tests with a concrete mix containing a processed coffee grounds husk with a microwave muffle furnace without any additional temperature controllers such as thermocouples placed inside of the specimens. 

### 4.2. Large-Scale Tests

These types of tests are usually restricted to a single event, which helps to determine the degree of fire performance produced by a selected structural composition [138]. There are already a few models regarding the speed of the temperature increase in evolution-like fire curves: HC, ISO834, RWS, RABT, and HC_inc_ [139], as shown in Figure 1.

Such factors as the maximum temperature, duration of fire exposure, time of decay, and heat release are crucial parameters to be considered in the fire testing of materials. The maximum temperature is the highest temperature reached during the test, and it is an important indicator of the material’s thermal resistance. The duration of fire exposure represents the time the material is exposed to fire, and it can affect the material’s structural integrity and mechanical properties. The time of decay is the time taken by the material to reach a certain temperature after the end of the fire exposure, and it can provide insight into the material’s cooling behavior. Lastly, the heat release is the total amount of heat released during the combustion process. It is an important factor in evaluating the fire hazard of a material. All these factors must be considered to accurately assess the fire performance of materials and ensure their safety in real-life fire situations. Nowadays, there are few developed custom-made investigations for specific types of concrete structures, especially underground objects where fire safety is a priority [140,141,142,143]. Yan et al. [140] constructed a firebrick chamber with a cover plate to conduct their test. Duan et al. [141] introduced a smoke chamber with a cooling tank for fumes. Factors such as the speed of the increase in temperature, presence and nature of the load, temperature thresholds, thermal conductivity, flammability, combustibility, spalling, cracking discolorations, changes in geometry, and many others are carefully observed, monitored, and analyzed to provide a full set of data. This type of setup arrangement is usually required to involve a team of specialists, consultants, and apparatuses. A reinforced concrete shield for metro-structure usage was tested following a standard ISO834 curve with a time duration of 45 and 90 min [140]. A furnace chamber made of firebrick was especially erected to proceed this test. K-type thermocouples and load transducers were installed to monitor and gather all data. Ring et al. [143] casted a 6 m long concrete frame which was placed in a customized heating chamber equipped with oil burners and ventilation chimneys. The temperature, geometry changes, and acoustic, crack, and spalling measurements were carried out to collect all the data. Large-scale tests are necessary to explore the fire-resistance field. There is a significant research gap regarding the behavior of sustainable concretes under high-temperature conditions, especially in large-scale testing. More data are needed to understand how these materials perform and to determine their potential use in fire-resistant structural elements.

## 5. Effects of Selected Factors

### 5.1. Effects of Aggegrates

It is well known that aggregates expand when exposed to elevated temperatures [144]. This event leads to the compression of the binder paste. As an outcome, the binder–aggregate bond is the most fragile element at a high temperature exposure. The major damage to concrete is caused by cracking, which is amplified by unequal thermal strains between the aggregates and the binder matrix. Aggregates occupy approximately 80% of the volume of concrete. There is no standardized classification system of aggregates regarding their behavior at elevated temperatures. The thermal stability of aggregates is characterized by physical and chemical stability upon exposure to elevated temperatures. It is defined by a series of thermal tests, i.e., dilatometric or thermogravimetric tests. A low thermal strains coefficient, negligible strains, low peaks over differential thermal analysis (DTS), or thermogravimetric analysis (TGA) curves are highly desirable. Thermal expansion properties differ for every mineral based on their mineralogical composition [106].

Aggregates are categorized into two types based on their mineralogy, i.e., siliceous (S) and calcareous (C) [145]. Eurocode 2 Part 1–2 and the majority of research in this field point out more benefits of using C types in a concrete mix [146,147]. As it is a processed natural stone, each batch of it might vary. In addition, it might have an impact on the fire performance. Thus, aggregates with an identical chemical composition could also perform differently in the same temperature increase scenario [148,149]. The usage of S-type aggregates leads to the most extensive changes and disturbance. The major change occurs due to a reversible crystalline transformation of the β − α inversion in quartz aggregates and sands at 573 °C. It results in a rapid volume increase, 1–5.7% [150]. The presence of water in the pores has a high impact on the fire resistance [151], similarly as the differential thermal stress [152] and presence of cryptocrystalline silica in aggregates [153]. The presence of flint results in increased cracking and spalling due to the high thermal instability of this material. The high vapor pressure accumulates in the laminar microstructure of flint [148,154]. The intergranular crazing leads to linked and clustered bigger cracks by shaped and regular cryptocrystalline silica in quartz crystals. Aggregates made from limestone and dolomite perform better due to the presence of calcium carbonate, which starts to decompose at 700 °C into lime (CaO) and reaches its peak at 800 °C and ends at 898 °C [147]. Rehydration of CaO may occur during cooling and cause a 44% expansion [46]. Exposure to increasing temperatures might result in reaching the melting point by the aggregates. The melting point of most minerals is above 1000 °C, i.e., granite 1210–1250 °C, basalt 1050 °C [155]. The size of aggregates plays a significant role in the formation of cracks and spalling. Aggregates smaller than 10 mm were observed to promote the formation of cracks and spalling, while larger aggregates tended to increase the resistance against these type of failures [16].

### 5.2. Effects of Fibers

The application of fibers in concrete can enhance its performance by an improvement in the durability and mechanical properties, lower shrinkage, or inducement of the ability to self-repair [156,157,158,159,160,161]. Fibers can limit cracking and spalling due to the thermal incompatibility of concrete ingredients. A supplementing geopolymer concrete mix with basalt fibers (1%vol) resulted in the improvement in the geometrical stability of a specimen regarding thermal stress and the incompatibility of a matrix up to temperatures of 1000 °C [162]. Any major changes regarding the flexural strength occurred in the range of 600 °C and 700 °C. However, over 700 °C, the flexural strength increased twice. It was presumed that this improvement is related to the sintering of the geopolymer matrix leading to the improvement in the fiber–matrix adhesion. No noticeable deterioration was observed in the fibers after being exposed to temperatures ranging from 600 °C to 1000 °C. Similarly, a reinforced metakaolin geopolymer with an alumina–silica–zirconia fiber appeared to prevent loss of strength due to thermal stress [163]. The optimum mix was obtained with a 1.0% volume of fibers. At 600 °C, the specimen showed a reduction in the compressive strength of up to 65% compared to the specimen tested at ambient temperatures, and 75% at 800 °C, respectively. Haddaji et al. [164] studied the effects of phosphate sludges on geopolymer concretes. Specimens were losing flexural strength when exposed to temperatures between 25 and 600 °C, but at higher temperatures, the flexural strength increased by 16.5%. Increased concrete production costs and partial carbon oxidation limit the usage of these fibers [162,165]. A geopolymer containing 53% vol of e-glass and 60% vol of basalt fibers exposed to 500 °C showed a 50% decrease in the initial flexural strength [166]. Basalt fibers were able to sustain 194 MPa of the flexural strength in samples exposed to 600 °C [167]. Furthermore, no visible cracks or delamination were observed. Similar trends were observed by Riberio et al. [168], who investigated a metakaolin-based geopolymer (reinforced K geopolymer composite (KGP)) with a basalt chopped strand mat (42 wt%)/KGP and woven fabric (47 wt%)/KGP. The basalt fabric showed better results. The flexural strength of woven fabric-reinforced composites was tested at high temperatures, revealing that at 300 °C, the flexural strength (17.8 MPa) decreased to almost half of the strength observed at room temperature (45.2 MPa). Nonetheless, at 600 °C, the flexural strength increased to 56.7 MPa, which was even higher than the strength at room temperature. DavoudianDehkordi et al. [169] have proven the enhanced properties of a lithium-based geopolymer with basalt fibers up to temperatures of 200 °C. The flexural strength of the specimen without fibers was 0.83 MPa, while the specimen containing fibers had a strength of 3.25 MPa. The havoc of e-glass fibers, debonding and decomposition of the matrix due to the expansion of fibers occurred at >600 °C in a metakaolin-based geopolymer activated by a mix of a sodium and potassium solution [170]. Whereas the flexural strength of a carbon-reinforced composite was improved from 17.8 to 55.8 MPa at 1000 °C. It was also reported that basalt fibers led to a complete transformation into a diffused ceramic net, which toughened the whole structure of the geopolymer at elevated temperatures. Geopolymer mortar mixes with PVA fibers showed a good performance in pre- and postheat exposure [171]. At 200 °C, the GP2 sample without any PVA fibers had a compressive strength of 27.1 MPa and 20.8 MPa at 400 °C. The geopolymer GP2F with fibers, on the other hand, had a compressive strength of 44.7 MPa at 200 °C and 27.8 MPa at 400 °C. The pull-out strength remained unchanged at 200 °C. The reports indicated that the temperature range of 300–400 °C is critical for BFC (biofibrous concrete) [172]. Nonetheless, incorporating lignocellulose fibers can help to reduce the occurrence of microcracks and explosive spalling in high-performance concrete and ultra-high-performance concrete subjected to high temperatures, in contrast to the unreinforced concrete composites. Specimens made of fiber-reinforced concrete were produced using 0.5% coconut and polypropylene fibers, and then subjected to temperatures ranging from 200 °C to 1000 °C [173]. Coconut fiber-reinforced cubes showed a superior performance compared to OPC cubes. Coconut fiber-reinforced concrete cubes exhibited a compressive strength of 15.42 MPa at a temperature of 400 °C and 8.04 MPa at 800 °C. The compressive strength for the OPC specimens was 14.13 MPa at 400 °C and 7.09 MPa at 800 °C. Netinger et al. [174] conducted a study on the impact of hemp fibers on the fire-resistant characteristics of concrete and the influence of the fiber treatment on the concrete’s strength properties. The assessment was carried out at two temperature levels (20 °C, 400 °C). The results indicated that incorporating hemp fibers into the concrete matrix slightly improved the initial compressive strength when compared to the control sample (approximately 2–12%). The compressive strength loss at 400 °C was approximately 43.43%. The incorporation of fibers into a geopolymer concrete matrix has countless potential in the development of a more sustainable and ecological material. In addition, it can contribute to smarter waste management by accommodating its volume and releasing storage areas and allowing for the creation of extended green zones.

### 5.3. Effects of Cooling Regime

The cooling regime of concrete after exposure to high temperatures is a crucial factor [175,176,177]. Rapid cooling can cause thermal shock and result in cracking, while slow cooling can lead to reduced strength and increased susceptibility to chemical attack [175]. Several cooling methods are used, including air cooling, water cooling, and using a special cooling compound or cooling tunnel. The choice of cooling method depends on several factors such as the type of concrete, the temperature of exposure, and the time of exposure. Additionally, the cooling rate can be controlled by adjusting the rate of cooling or the temperature of the cooling medium. Proper cooling can help maintain the structural integrity of concrete and prevent premature failure, making it an essential part of the overall design and testing process.

There are two common methods currently used [176]. In the first method, samples are left in the furnace until they reach room temperature which is far away from a real-case scenario. On the contrary, in the second method, samples are removed from the furnace and quenched in water [177]. This approach simulates well an actual structure, which is exposed to active fire-extinguishing systems. For example, the residual compressive strength of OPC with FA concrete specimens exposed to 800 °C and cooled by furnace cooling was 13% higher than the specimen cooled down by quenching. The sample exposed to 1000 °C showed a 0.9% higher residual compressive strength when cooled by furnace cooling than water quenching [175].

Others studied the effects of the cooling methods on a concrete mix containing fly ash and siliceous aggregate. The results showed that instant cooling influenced the concrete properties only when the sample was exposed to <330 °C [176]. Furnace cooling produced a 10% loss in the residual compressive strength, while cooling by quenching heated the specimen in water by 35% and 55%, respectively, when the sample stayed in water for 5 min and for 20 min. Thermal stress caused the formation of microcracks, which eventually led to a failure of the matrix. Mendes et al. [46] compared the effects of cooling methods on OPC and ecological concretes containing GGBFS exposed to temperatures of 400 °C and 800 °C. Tests have shown that, 24 h after exposure to 800 °C, the residual compressive strength loss for concrete with only OPC was 14%, while for the GGBFS blends it was 4–5% in a comparison of the specimen cooled down by a furnace and water quenching, respectively. After 7 days of cooling within a furnace after exposure to 800 °C, all specimens showed a further decline in their performance. The damage caused by the immediate water quenching was similar to the damage observed after 1 week of furnace cooling and exposure to the ambient humidity. This aspect should be considered when assigning the postfire damage of structures.

The experiments demonstrated that exposure to ambient moisture resulted in the rehydration of CaO to Ca(OH)_2_, and the level of disorientation of the concrete samples was similar between water cooling and exposure to ambient humidity for one week. This allowed for the identification of two ranges of disorientation of concrete, with and without dehydration/rehydration of Ca(OH)_2_. These tests were supplemented with SEM and IR studies. The cooling process is a critical stage in the postfire assessment of concrete structures, as it can significantly affect the residual properties of the material. However, there is a lack of comprehensive studies that investigate the effects of different cooling methods on the mechanical and physical properties of blended cement and ecological concrete. In order to develop more resilient and sustainable concrete structures, it is crucial to gain a deeper understanding of how cooling methods impact the behavior of these materials under fire exposure. This requires more research and experimentation to identify the optimal cooling techniques and parameters for different types of blended cement and ecological concrete mixtures.

### 5.4. Heat Transfer Characteristic and Thermal Conductivity

The aspect of heat transfer and thermal conductivity of concrete are important factors to be considered when investigating fire damage mechanisms [178,179,180,181,182,183]. The correlation between the type of aggregates and the moisture content in concrete is a crucial dependency. Bentz et al. [180] have reported that siliceous and limestone aggregates have a major impact on thermal conductivity.

In addition, the transient plane method can be applied to evaluate the thermal performance of high-volume fly ash (HVFA) mortars and concretes at ambient temperatures. The Transient Plane Source (TPS) method is a technique used to measure the thermal conductivity and thermal diffusivity of different materials. It involves inserting a small, flat sensor between two specimens of the material being tested and applying a constant heat flux to one side. The sensor reads the temperature increase on the other side of the sample and calculates the thermal properties based on the rate of temperature alteration over time [184].

HVFA concrete with reduced thermal dynamism is characterized by a lower cost of utility cooling and heating in commercial and residential structures. The determination of the moisture resistance factor and thermal conductivity decrease with rising temperatures was reported by Kizilkanat et al. [179]. Li et al. [178] stated that the free water content has an impact on the thermal conductivity of an early age concrete. There are several models which can be used to estimate the thermal conductivity of undamaged and damaged concrete, including, for example, the mesoscale model formulated by Zhang et al. [185]. Insulation (I), integrity (E), and resistance (R) are three constituent factors of standardized fire test methods. First, two (I) and (E) are related to the performance of a system or element of separate behavior in contact with fire. Integrity (E) stands for a resistance of spread of flame or smoke to the nonexposed side of a structure. The ability to increase the temperature of the separated and nonexposed side of an element is defined by insulation (I). The characteristic of heat transfer is based on those two factors. It was concluded that to ensure safety and prevent hazards, the temperature variations should be within an acceptable range of up to 140 °C above the original temperature, with the maximum temperature at any point not exceeding 180 °C [186].

As previously demonstrated, it was generally observed that mixes with a 70 weight percentage of ground granulated blast-furnace slag (GGBFS) had an improved performance in terms of fire resistance at high temperatures [186]. The thickness of the concrete cover can be reduced by 5%. Furthermore, by applying the simplified method described in I.S.EN 1992-1-3 [187], the thickness can even be reduced to 10%. Due to the higher density of tested concrete samples containing GGBFS, the heat transfer properties were improved. Mortars based on OPC but containing granulated glass added as a fine aggregate showed an improved thermal conductivity in comparison with mixes containing an ordinary sand [188]. Others observed that the thermal conductivity of fly ash concrete increased due to the increase in the microenvironment relative humidity [189]. When it reached 100%, the thermal conductivity rose by about 22%. Fly ash concrete performed better when exposed to higher temperatures than OPC.

Higher temperatures influenced the acceleration of heat absorption. The thermal conductivity of concrete with a 30% fly ash replacement was found to be lower than that of ordinary concrete under identical conditions. One of the biggest problems is a lack of standardized methodology to define the thermal properties of concrete. One method which could be applied is the Transient Plane Source (TPS) technique. It allows for carrying out a test regarding the heat transfer in ambient and at elevated temperature conditions [190]. Due to the high cost and difficult access, it has not become popular yet.

## 6. Discussion

The mechanical properties and durability of eco-concretes are, in some cases, compromised when exposed to fire. Fire-related damage of concretes is usually assessed using small-scale indirect tests. These tests are not sufficient to provide the full set of data needed for modeling. The performance of eco-concrete is always reduced after exposure to elevated temperatures. Eco-concrete has different temperature thresholds depending on its composition. Color alteration, cracks, delamination, and spalling are very often the accompanying events which can be easily identified. Partial replacement of Portland cement by alternative binders and secondary cementitious materials also enables, in some cases, an enhancement in their performance when exposed to fire [63,125,129]. Additionally, certain fibers have shown an ability to reduce cracking and spalling of concretes [164,191]. An increased density of geopolymers helped to withstand better exposure to variable temperature gradients [125,192,193]. There are already a few well-known and respected standards for large-scale testing [139], but the variety and lack of quality control over small-scale testing create questions about replicability and reliability of the obtained results. During writing this paper, some factors influencing the fire performance of eco-concrete could be highlighted, as shown in Table 7. Those can be categorized into three main groups: composition-related, fresh concrete state-related, and exposure-related. The composition-related factors include the type and proportion of eco-friendly materials used in the concrete mix, the curing process, and the strength of the concrete matrix. The exposure-related factors include the duration and temperature of the fire, as well as the type of load and the cooling conditions that the concrete is subjected to after exposure to fire. The thermal conductivity and postfire behavior of eco-concrete has not been clearly understood.

GGBFS and FA have gained popularity as research subjects for material behavior under elevated temperatures due to their wide availability and easy accessibility. They have shown promising results as binder replacements in cement blends and in alkali-activated systems, as evidenced in Table 8. They visibly discolor under elevated temperatures, which could be very helpful in a postfire nondestructive structure assessment and the spalling event is not occurring. Cracks are introduced as a microcracks net, which allows water to migrate. Due to the formation of a new gel under elevated temperatures, their matrix is denser, more impacted, and robust compared to OPC. Their mechanical performance is superior compared to the OPC specimen. Regarding exposure to temperatures higher than 800 °C, all mixes fail, and the matrix disintegrates. It could be improved and postponed over time by the presence of fibers.

The performance of eco-concrete is influenced by a range of factors, and its behavior during a fire event is particularly difficult to predict due to the complex nature of the material. Therefore, it is essential to evaluate the various stages of the eco-concrete life cycle, including laboratory tests, casting, service life, fire events, and postfire scenarios, to ensure its optimal performance. All these stages play a critical role in the final performance of eco-concrete, and their thorough evaluation is necessary for a reliable and effective fire-performance assessment.

## 7. Conclusions

It has been demonstrated that concrete can be made more environmentally friendly by improving its fire resistance and ability to withstand high temperatures [201,202]. This can be achieved by using industrial waste materials, such as fly ash, ground granulated blast-furnace slag, rice husk, and mine tailings, used as binders or aggregates. The compressive strength of eco-concrete is generally well maintained when exposed to elevated temperatures. Between 600 °C and 700 °C, there are no major changes. Due to densification of the matrix, the compressive strength is often improved. However, at temperatures above 700 °C, a decrease in the compressive strength was reported. This reduction can be attributed to the thermal decomposition, thermal incompatibility of components, and the loss of moisture content within the binder matrix.

A sintering process refers to the bonding and densification of the particles at high temperatures, leading to the formation of a stronger and more cohesive structure. In many cases, the sintering event of the eco-concrete matrix resulted in an improved adhesion between the fibers and matrix. As a consequence, the overall mechanical properties were enhanced in comparison with OPC concrete.

Ecological concrete generally showed reduced spalling and less cracking compared to OPC concrete when exposed to elevated temperatures. The visible color alteration after exposure to fire does not necessarily indicate a compromise in the structural integrity or performance of the material. It could be a good indicator for nondestructive methods.

It was reported that some types of fibers did not show any noticeable signs of deterioration. The favorable effect of crack and spalling prevention was observed. It was assumed that fibers were able to withstand thermal stress and maintained their integrity. Some also indicated the induction of a self-healing process in fire-damaged samples.

The postfire scenario and cooling process has a significant impact on the properties and behavior of ecological concrete. Rapid cooling can cause high thermal stress and potential cracking due to the differential volume change in its components.

## 8. Future Research

Summarizing, there is still a lack of reliable and replicable data related to the performance of eco-concretes in fire. The following research needs could be indicated:Nanoscale research to fully understand the mechanism of influence of elevated temperatures on an eco-concrete element;An investigation about heating and the cooling rate and its influence on ecological and sustainable concrete performance. In addition, the postfire behavior of eco-concrete must be studied more profoundly;A proper approach and methodology to formulate a user-friendly concrete mix design with a minimized presence of hazardous materials and alkali solutions which can withstand high temperatures;Evidence that admixtures do not change the concrete properties under elevated temperatures;The bonding properties of different materials, such as reinforcement, in ecological concrete;A standardized method and methodology to carry out small-scale testing of specimens;Large-scale fire tests on ecological concrete. It remains unknown how the extremal factors such as the shape of the structure, wind, snow, applied loads, additional elements located in concrete structures, and fire-extinguishing systems (passive and active) could disturb the properties and overall performance of concrete elements;A developed and classified database of sustainable concrete components and possible configurations;A database, building codes, fire regulations, design and material approach, and user standards in a variety of scenarios, situations, and timelines;A life-cycle assessment analysis and a determination of the potential reusage of recycled concrete to minimalize the volume of waste;Digitalization, collaborations, and open access combined with artificial intelligences.

## Figures and Tables

**Figure 1 materials-16-04212-f001:**
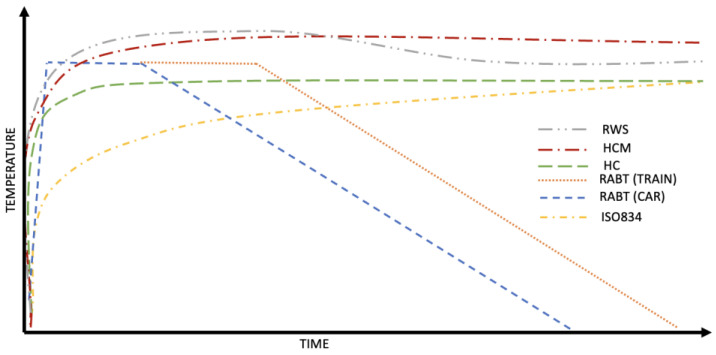
Adapted schematic comparison of heating curves.

**Table 2 materials-16-04212-t002:** Ordinary Portland Cement (OPC) phase decomposition [39,40,41].

Temperature	Phase Changes
>200 °C	Part of the ettringite undergoes dehydration, as does the calcium silicate hydrate (C–S–H).
>300 °C	The dehydration of ettringite and subsequent dehydration of calcium silicate hydrates (C–S–H) occur.
>500 °C	The subsequent dehydration of calcium silicate hydrates (C–S–H) and the conversion of portlandite Ca(OH)_2_ to dihydrate form.
>800 °C	The continued dehydration of calcium silicate hydrates (C–S–H) and the partial decomposition of carbonate.
>1200 °C	The temperature at which aggregates melt.

**Table 4 materials-16-04212-t004:** Fly ash (FA) concrete phase decomposition [67].

Temperature	Phase Changes
>200 °C	Free water evaporation, mass loss, enhanced mechanical properties by temperature-induced gel formation, fewer unreacted particles, increased density due to filled-up pores by newly formed gel.
200–400 °C	High contraction which leads to removal of chemically bound water, continuation of gel formation and increasing the density, peak strength value [55].
400–600 °C	Slow shrinkage caused by dihydroxylation (formation of hematite, mullite) [17], strength loss due to crystallization of gel.
600–800 °C	Further shrinkage, oxidation of present iron oxides, decrease in porosity, formation of micropores and microcracks [29].
800–1000 °C	Final part of particles coalescence, deterioration of matrix, critical damage of element.

**Table 5 materials-16-04212-t005:** Calcium sulfoaluminate (CSA) phase decomposition [39].

Temperature	Phase Changes
<90 °C	Ettringite dehydration and decomposition to monosulfite and calcium sulfate.
>150 °C	Partially monosulfite dehydration.
200–230 °C	Alumina trihydrate dihydroxylation.
>450 °C	Dehydration of monosulfite.

**Table 6 materials-16-04212-t006:** Adapted data from eco-concrete testing in elevated temperatures [29].

Temperature [°C]	OPC [MPa]	Eco-Concrete [MPa]
Ambient	35	65
200	32	55
400	30	52
600	17	58
800	9	54

**Table 7 materials-16-04212-t007:** Factors influencing fire resistance of eco-concrete.

Category	Parameters
Composition	Binder type and amount [56], gradation, shape, and amount of coarse aggregate [102], water to binder ratio [18,192,193], amount and type of additives [97].
Fresh concrete properties	Concrete density [18,192,193], curing regime [194].
Elevated temperature exposure	Scale of tests (small or large scale) [131,141,143], heating rate [38,175], cooling rate and type [175], fire flame vs. temperature rise [147], geometry and size of element [21], spalling and cracks [22,35,81], surrounding conditions [46,126], applied loads [21,36,53,171], additional events [141,143]

**Table 8 materials-16-04212-t008:** Selected examples of eco-concrete in elevated temperature conditions.

Type of Binder	Time [min]	Compressive Strength		Other Events		Ref.
		400 °C	800 °C	400 °C	800 °C	[195]
**OPC + GGBFS (fine aggregate) (100%)**	60	Ref: 51.60 MPa	Ref.: 9.4 MPa	Discoloration	Discoloration, cracks	
		58.6 MPa	18.1 MPa	Discoloration	Discoloration, cracks	
**OPC 42.5** **+ FA (70 wt%)**	120	Ref.: 98.93%	Ref.: 45.47%	Mass loss,	Decomposition of matrix, microcracks	[196]
		289.86	91.63%	Mass loss, increased density, better mech. performance	Decomposition of matrix	
**FA + GGBFS (60 wt%)**	60	Ref.: N/A	Ref.: ≈51.00 MPa	N/A	Higher shrinkage, porosity, lower density	[118]
		N/A	≈66.00 MPa	N/A	Lower shrinkage, porosity, higher density	
**OPC (70 wt%) + GGBFS (30 wt%)**	60	70/30 32.70 MPa	70/30 28.4 MPa	Lower mass loss, micro cracks	Cracks, higher mass loss, loose structure	[197]
		30/70 7.2 MPa	30/70 6.00 MPa	Higher mass loss	Cracks, lower mass loss, loose structure	
**OPC 50 (wt%) + GGBFS (50 wt%)**	≈200	Ref.: ≈80%	Ref.: ≈35%	Rehydration of CaO into CaOH2	Rehydration of CaO into CaOH2, expansion of matrix	[46]
		≈75%	≈27%	Rehydration of CaO into CaOH2	Rehydration of CaO into CaOH2, expansion of matrix	
**OPC + FA** **(50 wt%)**	420	Ref.: ≈22 MPa	Ref.: ≈4 MPa	No cracks	Cracks on edge of specimen, higher mass loss	[198]
		≈33 MPa	≈10 MPa	No cracks	Cracks on edge of specimen, lower mass loss	
**FA+ RH**	N/A	Ref.: (heating curve, after 30 min) 10.5 MPa	Ref.: (heating curve, after 60 min) 4.9 MPa	Color alteration, spalling after 90 min		[194]
		(heating curve, after 30 min) 10.3 MPa	(heating curve, after 60 min) 4.2 MPa	Color alteration, spalling after 90 min		
**OPC+ FA** **(50 wt%)**	420	Mix II ≈33 MPa	Mix II ≈8 MPa	Color alteration	Cracks, mass loss	[199]
		Mix IV ≈55 MPa	Mix IV ≈15 MPa	Color alteration	Cracks, mass loss	
**FA + granite waste aggregate (50 wt%)**	N/A	(heating curve after 30 min) Ref. SSD: 13.4 MPa	(heating curve, after 60 min) Ref. SSD: 4.1 MPa	Red tones, cracking, or spalling after 90 min, increased thermal conductivity		[107]
		13.6 MPa	4.3 MPa	Red tones, cracking, or spalling after 90 min, increased thermal conductivity		
**FA + GGBFS (50%)**	120	Ref.: ≈140%	Ref.: ≈70%	No cracks	Orange tones, higher recovery performance	[200]
		≈81%	≈28%	Red tones, higher mass loss, micro cracks	Pink tones, higher mass loss, cracks, intensive shrinkage	
**FA+ GGBFS (100%, 50/50%, 100 wt%) A: Na 8% (1.5)**	60	N/A	100% FA ≈3 MPa	N/A	Orange tones, higher, cracks	[30]
		N/A	50/50% FA ≈5 MPa	N/A	Dark red tones, no cracks	
		N/A	100% S ≈2 MPa	N/A	Orange red tones, cracks	

## Data Availability

No new data were created or analyzed in this study. Data sharing is not applicable to this article.

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
