# Peer review of "Eco-Concrete in High Temperatures"

_materials, 2023, doi:10.3390/ma16124212_

Round 1
Reviewer 1 Report
(1) The abbreviations should be given full names for their first use, e.g., GGBFS and OPC-based in abstract section, as well as in other sections.
(2) The abstract section could be re-organized. The description of the eco-concrete accounts takes too much space, while the contents beyond the general mechanism of eco-concrete upon fire and high temperatures scenarios is rarely discussed, and the latter should be the most important, and is the main objective of this manuscript.
(3) In section 2.2, Fly ash (HA) concrete, in which HA should be FA.
(4) In section 5, there are two sections 5.2, and the second one should be 5.4.
(5) In section 4, whether there is any simulation method used to study the fire resistance of concrete materials?
(6) The conclusions are too concluded, and there are too many further to-dos. As a review paper, to point out further research directions is good and necessary, but the conclusions of your review should be given in details at first.
The manusript is well-written in language.
Reviewer 2 Report
I would suggest that sections and sub-sections of the manuscript be divided into paragraphs. Each paragraph should convey a single message / type of information pertaining to the section it belongs to.
Please check the manuscript for typing inconsistencies, such as the Celsius degrees.
Line 223 - please do no use acronyms before giving full name (CSA)
Lines 225-227 - please rephrase using shorter sentences. The message the authors are trying to convey, although scientifically valid, is hard to grasp
Table 5 - was the same phenomenon reported for temperature range 200-230 and higher than 450? This information is not consistent with line 232.
Lines 316-324 - while the presented information is valuable, I would recommend the authors compile it and present it by means of graphs as they are easier to understand than following long paragraphs of text.
Line 382-384 - please use shorter sentences.
Figures 1, 2, 4 and 5 - please make sure you have the permissions for reprints. Citing the source is not enough.
Figure 3 - is this authors' own work? If so, please provide values on the two axes and the units of measure
Line 528 - how do the presence of fibers enhance the hydration process of cement?
Line 548 - "showed over 50% of the ..."? 50% increase / decrease?
Line 552 - please give full description of KGP
Line 567 - the readers do not know the details of [180] research. Therefore, GP2 sample bears no significance. Please describe the main features of that GP2.
Lines 662-664 - the statement is ambiguous and confusing.
The Conclusions section is rather general and misses the some of the observations highlighted in the manuscript. This section should summarize the findings of the state-of-the art in a concise and clear manner. Maybe one or two paragraphs should elaborate on future research directions or what needs to be done.
Line 19 - "fire-resistant aggregates"
Line 19 - what methodology?
Line 20 - "total or partial..."
Lines 101-102 - the sentence is unfinished
Line 108 - "loss of strength..."
Lines 109-110 - please revise and watch for punctuation
Line 124 - "in the building..."
Line 130 - used as a pozzolanic (same comment for line 132)
Line 131 - please replace "consumer" by "consumes"
Line 155 - please remove "of". What do you understand by "unparallel"? Are those cracks inclined or vertical?
Line 166 - the sentence is not clear. Please revise.
Line 187 - FA instead of HA
Line 191 - please remove "that"
Line 197 - fire curve
Line 204 - "tested" instead of "testes"
Line 232 - the sentence is not finished
Lines 240-242 - please rephrase using shorter sentences
Line 270 - please replace "its" by "their"
Line 277 - "rice" instead of "rise"
Line 362 - "due to the absence of organic particles, it has been applied...."
Line 455 - "are crucial parameters to be considered..."
Line 475 - "was especially"
Line 477 - "placed in a customized"
Line 504 - "calcareous" instead of "cancerous"
Line 512 - "the presence of water in the pores..."
Line 577 - "showed superior / enhanced performance"
Round 2
Reviewer 1 Report
The authors addressed my problems, the manuscript is well-written now, and it is suggested to be accepted.
Reviewer 2 Report
Line 290 - "their" instead of "thier"
Line 644 - consider as a starting of a new paragraph (presenting the 2 cooling methods)
Like 658 - could be the start of another paragraph (effect of cooling methods on eco concrete)
Line 672 - could be the start of another paragraph
Line 701 - could be the start of another paragraph
Line 727 - could be the start of another paragraph
